# Immune Cell Density Evaluation Improves the Prognostic Values of Staging and p16 in Oropharyngeal Cancer

**DOI:** 10.3390/cancers14225560

**Published:** 2022-11-12

**Authors:** Géraldine Descamps, Sonia Furgiuele, Nour Mhaidly, Fabrice Journe, Sven Saussez

**Affiliations:** 1Department of Human Anatomy and Experimental Oncology, Faculty of Medicine, Research Institute for Health Sciences and Technology, University of Mons (UMONS), Avenue du Champ de Mars, 8, B7000 Mons, Belgium; 2Laboratory of Clinical and Experimental Oncology, Institute Jules Bordet, Université Libre de Bruxelles (ULB), Rue Meylemeersch, 90, B1070 Anderlecht, Belgium; 3Department of Otolaryngology and Head and Neck Surgery, CHU Saint-Pierre, Rue aux Laines, 105, B1000 Brussels, Belgium

**Keywords:** oropharyngeal cancer, p16, staging, TNM, prognostic, T regulatory lymphocytes, FoxP3, Langerhans cells, CD1a, immune cells

## Abstract

**Simple Summary:**

Human papillomavirus (HPV) has become the major risk factor for the development of oropharyngeal squamous cell carcinomas (OPSCCs), the incidence of which continues to grow in Western countries. Their biological features, associated with a better prognosis as well as a greater response to treatment, has already led to their staging system reclassification and to the development of clinical trials to deintensify the therapeutic approaches. In this context, we proposed to evaluate the recruitment levels of immune cells to ameliorate the classification of some groups of OPSCCs that are always associated with poor outcomes. For this purpose, we scored the density of CD8 and FoxP3 lymphocytes, CD68 macrophages and CD1a Langerhans cells and associated the significant cells with either p16 status or TNM staging to create strong combinations that demonstrated powerful prognostic values in such patients. These results encourage the development of further studies based on the inclusion of immune criteria in the classification of OPSCCs.

**Abstract:**

The incidence of oropharyngeal cancers (OPSCCs) has continued to rise over the years, mainly due to human papillomavirus (HPV) infection. Although they were newly reclassified in the last TNM staging system, some groups still relapse and have poor prognoses. Based on their implication in oncogenesis, we investigated the density of cytotoxic and regulatory T cells, macrophages, and Langerhans cells in relation to p16 status, staging and survival of patients. Biopsies from 194 OPSCCs were analyzed for HPV by RT-qPCR and for p16 by immunohistochemistry, while CD8, FoxP3, CD68 and CD1a immunolabeling was performed in stromal (ST) and intratumoral (IT) compartments to establish optimal cutoff values for overall survival (OS). High levels of FoxP3 IT and CD1a ST positively correlated with OS and were observed in p16-positive and low-stage patients, respectively. Then, their associations with p16 and TNM were more efficient than the clinical parameters alone in describing patient survival. Using multivariate analyses, we demonstrated that the respective combination of FoxP3 or CD1a with p16 status or staging was an independent prognostic marker improving the outcome of OPSCC patients. These two combinations are significant prognostic signatures that may eventually be included in the staging stratification system to develop personalized treatment approaches.

## 1. Introduction

Among head and neck squamous cell carcinomas (HNSCCs), those affecting the oropharynx arise due to human papillomavirus (HPV) infection or the influence of traditional risk factors [1]. Recently, HPV+ oropharyngeal cancers (OPSCCs) have been included in the 8th edition of the American Joint Committee on Cancer (AJCC) to re-evaluate their classification based on their better prognosis and unique treatment regimens [2]. The reasons for these longer survival times are mostly due to clinical and molecular factors. Indeed, the profile of affected patients corresponds to younger and nonsmoking males who better respond to conventional treatments. Additionally, these cancers can often originate from the epithelium of the tonsil crypts, which are rich in immune cells, favoring local immune activation and immune surveillance [3]. For several years, the involvement of the immune system in tumor progression has been well recognized, so it appears necessary to characterize this defense environment by specifying the expression profiles of immune biomarkers. This characterization will aim to improve the therapeutic efficacy and decrease the intensity of OPSCC treatments.

In this context, the most-studied immune type is undoubtedly CD8+ cytotoxic T lymphocytes, which are the major defensive elements against tumor cells. Their massive infiltration within the tumor generally predicts better survival [4,5,6,7]. Regarding HPV, some studies demonstrated that T-cell infiltration was associated with a good prognosis regardless of HPV status [8,9], while others demonstrated that higher CD8+ T-cell density was positively related to HPV status [10]. Regulatory T cells (Tregs) are also important actors in the immune tumor microenvironment (TME) of OPSCCs. They are characterized by the expression of the forkhead transcription factor (FoxP3), which is a key regulator of their functions and is used in most studies to distinguish Tregs from other immune cells [11,12,13]. Although they are physiologically involved in immune tolerance, their roles and impacts in the context of cancer are still quite controversial. Their involvement is demonstrated in the inhibition of antitumor responses leading to immune escape. In different types of cancers, such as gastric, hepatic, breast and melanomas, their high density is often reported to be associated with a poorer prognosis, whereas their presence in head and neck and colon cancers is synonymous with a better outcome [14]. Indeed, we have previously demonstrated that higher Treg infiltration correlated with longer overall survival (OS) and recurrence-free survival (RFS) in HNSCC patients [15,16].

Macrophages and particularly M2 tumor-associated macrophages (TAMs) with protumor effects are responsible for Treg differentiation and are able to create an immunosuppressive environment favoring tumor growth, notably through the secretion of cytokines (IL-10, TGFβ, TNFα) [17,18]. Since HNSCC is largely infiltrated by TAMs (up to 30%), their expression and abundance are often related to a poor prognosis and to the occurrence of recurrence [19,20,21,22]. In addition, our previous study demonstrated that the infiltration of CD68+ cells into the tumor increased during tumor progression and that a high infiltration of such macrophages correlated with shorter survival. In relation to HPV status, we previously observed that macrophage recruitment was higher in HPV+ tumors than in HPV- tumors [23]. Indeed, HPV is able to modulate the TME to promote tumor immune escape. In this respect, we also observed that the number of Tregs was increased in HPV+ HNSCCs and that, conversely, Langerhans cell (LC) infiltration was significantly decreased in these patients [15,24]. Of note, this population of dendritic cells specializes in presenting antigens to T cells, including viral antigens, to generate immune defenses against HPV. The modulation of the immune system by HPV has also been demonstrated by Nguyen et al., who reported a decrease in the number of LCs in the OPSCC stroma of infected patients [25]. Given the heterogeneity of OPSCC, some tumor regions are infiltrated by immune cells, reflecting different clinical outcomes. Indeed, intratumoral and stromal drivers may differentially influence the evolution of the cancer, leading either to tumor progression or regression, highlighting the importance of considering each compartment separately [26,27].

Currently, the OPSCC classification remains based on TNM clinical parameters assessing tumor extension, lymph node involvement and distant metastasis presence. In an era where immunoscores are becoming increasingly relevant and widespread, it appears that categorization based on HPV status or TNM alone is undervalued. Recently, an immunoscore assessing tumor-infiltrating lymphocytes (TILs) in colon cancer has been established as a new classification model, which has demonstrated better prognostic prediction than the classical TNM system [28,29]. Similarly, we recently identified a three-marker-based immunoscore that had a stronger prognostic performance than tumor stage [30]. Quantification of immune cells appears to be a promising approach but requires a comprehensive investigation of the immune landscape in OPSCCs. Therefore, in this study, we quantified the expression of CD8, FoxP3, CD68 and CD1a, compared their recruitment according to clinical characteristics and determined their prognostic value in a series of patients with oropharyngeal cancers. The aim of this study was to improve the classification system of OPSCC patients, regardless of HPV status, and to investigate the extent to which a combination of immune cells and clinical variables influence prognosis.

## 2. Materials and Methods

### 2.1. Patients and Clinical Characteristics

Patients with pathologist-confirmed oropharyngeal cancer were selected based on former cohorts and were diagnosed between 2001 and 2021. The 194 formalin-fixed, paraffin-embedded (FFPE) tumors were derived from patients who had undergone curative surgery at CHU Saint-Pierre (Bruxelles, Belgium), Jules Bordet Institute (Bruxelles, Belgium) and EpiCURA Baudour Hospital (Baudour, Belgium). Institutional research ethics board approvals were obtained, and written informed consent was signed by each patient enrolled in this retrospective study (Jules Bordet Institute, number CE2319). The cohort of patients and the clinicopathological data are summarized in Table 1.

### 2.2. DNA Extraction

DNA extraction from FFPE specimens was performed as described in our previous publications [31]. Briefly, sections were deparaffinized and digested with proteinase K by overnight incubation at 56 °C. DNA was purified using the QIAmp FFPE tissue kit according to the manufacturer’s protocol.

### 2.3. Detection of HPV by Polymerase Chain Reaction (PCR) Amplification

GP5+/GP6+ primers were used to amplify a consensus region located in the L1 region of the HPV genome. This PCR protocol chosen to detect HPV DNA has been fully described in a previous publication [31].

### 2.4. Real-Time PCR Amplification of HPV Type-Specific DNA

All DNA extracts were tested at the Algemeen Medisch Laboratorium (Antwerp, Belgium) for the presence of 18 different HPV genotypes using TaqMan-based real-time quantitative PCR that targeted type-specific sequences of the viral genes, as previously described [31].

### 2.5. p16 Immunohistochemistry

To determine the transcriptional activity of HPV, each HPV-positive case was further immunohistochemically evaluated for p16 expression using a mouse monoclonal antibody (CINtec p16, Ventana, Tucson, AZ, USA) and an automated immunostainer at the Jules Bordet Institute (Bond-Max, Leica Microsystems, Wetzlar, Germany). Briefly, after epitope retrieval (pH 6), sections were incubated with the p16 antibody for 30 min. Then, polymer detection was performed using Bond Polymer Refine Detection according to the manufacturer’s protocol (Leica, Wetzlar, Germany), and the slides were counterstained with hematoxylin and Luxol fast blue. Tissue sections from cervical lesions were used as positive controls. A negative control was performed by omitting the primary antibody. Tumors were considered positive when strong and diffuse staining was scored both in the nucleus and the cytoplasm and in ≥70% of the tumor.

### 2.6. Evaluation of Immune Cell Recruitment by Immunohistochemistry

Immunohistochemistry, targeting immune cells, was performed on 5 µm deparaffinized and alcohol-rehydrated tissue sections. The peroxidase activity was saturated with H_2_O_2_ for 10 min followed by antigen retrieval in EDTA/H_2_O or citrate buffer/H_2_O (see Appendix A). Tissues were incubated with casein 0.5% for 1 h to block nonspecific epitopes and then with the specific primary antibody for 1 h at room temperature (RT) or overnight at 4 °C as described in Appendix A. Finally, the samples were incubated with BrightVision Poly-HRP–IgG (Klinipath, Duiven, Holland), and the antigens were visualized by the addition of a solution of 3–3′ diaminobenzidine and H_2_O_2_ buffer (Liquid DAB, San Ramon, CA, USA) before counterstaining with Mayer’s hemalun and Luxol fast blue. Tonsil tissues from healthy patients were used as positive and negative controls. The number of each immune cell type was counted in 5 randomly selected fields at 400× magnification by three investigators (G.D., S.F., N.M.) in both stroma (ST) and intratumoral (IT) compartments. The mean was calculated for each patient and normalized to a 1-mm^2^ area. Finally, optimal cutoffs allowing the best separation between low- and high-expressing groups of each immune cell type by compartment were calculated.

### 2.7. Statistical Analyses

Statistical analyses were performed using SPSS software version 21 (IBM, Portsmouth, UK). Univariate Cox regression analyses were performed to identify prognostic variables influencing OS as well as to calculate hazard ratios (HRs), 95% confidence intervals and significance. Multivariate analyses were applied to assess the independent contributions of immune cells to OS in the presence of other covariates, including p16 status and tumor stage. Kaplan–Meier curves were successively assessed for OS. The prognostic value of immune markers related to OS was evaluated based on the calculated optimal cutoffs. Immune cells expressed in different subgroups (p16− vs. p16+, low stages vs. high stages) were compared using the nonparametric Mann–Whitney U test. In all cases, two-sided *p* values < 0.05 were considered statistically significant.

## 3. Results

### 3.1. Immune Cell Density and Patient Survival

The typical immunohistochemical expression of cytotoxic T-lymphocytes, regulatory T-lymphocytes, Langerhans cells and macrophages was evaluated using specific antibodies against CD8, FoxP3, CD1a and CD68, respectively (Figure 1). Their quantitative expression was assessed in both stromal (ST) and intratumoral (IT) compartments by counting their number in five random fields, and their density was defined as low or high in each compartment based on optimal cutoffs evaluated regarding the *p* values for OS of patients. The cutoff values were 718 cells/mm^2^ (CD8, ST), 110 cells/mm^2^ (CD8, IT), 552 cells/mm^2^ (FoxP3, ST), 83 cells/mm^2^ (FoxP3, IT), 61 cells/mm^2^ (CD1a, ST), 138 cells/mm^2^ (CD1a, IT), 293 cells/mm^2^ (CD68, ST) and 188 cells/mm^2^ (CD68, IT).

Among the 194 surgical specimens of OPSCC, 44 could be analyzed for CD8 expression, 77 for CD68, 66 for CD1a and 69 for FoxP3 (Table 2). Their distribution was then compared between different subgroups of interest based on the median to display the variability of their density between p16-positive and p16-negative patients and between low- and high-stage tumors (Table 2).

Then, the associations between OS and the eight immune variables were also assessed to identify the immune type most likely to positively impact patient survival. Univariate Cox regression analysis revealed that three factors were significantly associated with OS, namely, CD68 IT, FoxP3 IT and CD1a ST (Table 3).

Moreover, to identify the best combination for improving p16 status and/or staging, a comparison of these two variables by Mann–Whitney tests according to immune cell density was performed. The results showed that CD68 infiltration in the ST is significantly associated with p16 status as well as IT Treg infiltration. Additionally, we observed that the density of Langerhans cells in the ST was significantly associated with the tumor stage of patients (Table 4).

Representative box plots of significant Mann–Whitney tests comparing the cell density of CD68 and FoxP3 between p16+ and p16− patients revealed a lower density of macrophages in the ST (*p* = 0.005) along with a higher proportion of Tregs in the IT (*p* = 0.02) compartments of p16-positive tumors (Figure 2). Among these two immune cell types, only FoxP3+ cells correlated with patient survival (Table 3), supporting further investigation of the potential impact of their combination with p16 status. In the same manner, only CD1a ST had prognostic value (Table 3) and a correlation close to significance with patient staging (Table 4), leading to the examination of such a combination regarding OS.

### 3.2. Combination of p16 Status and Regulatory T-Lymphocyte Density and Correlation with Patient Survival

Among the 194 patients included in this study, 52 had positive expression of p16 corresponding to a transcriptionally active infection. Regarding the relationship between p16 expression and the survival of OPSCC patients, p16-positive status predicted a significantly better prognosis (*p* = 0.01, Figure 3A). Next, we determined whether there may be a relationship between the density of FoxP3 in IT and the OS of OPSCCs, and the results showed that patients with high levels of FoxP3 had significantly longer survival than those with low numbers of FoxP3+ cells (*p* = 0.018, Figure 3B and Table 3).

Based on these significant correlations between OS and p16 status and OS and Treg infiltration, combinations of p16 and Treg were established, and four survival curves were plotted to determine the prognostic performance of each subgroup (Figure 3C), as summarized in Table 5. Finally, the two subgroups of patients who were associated with better survival were pooled to create a score (high versus low) based on p16 expression and Treg density. The results showed that OPSCC patients with a high score had significantly better survival than those with a low score (*p* = 0.012, Figure 3D). This score provides a stronger separation of patients than p16 and Treg status alone.

### 3.3. Combination of Staging and Langerhans Cell Density and Correlation with Patient Survival

Next, we investigated the relevance of the CD1a marker and staging, alone or in combination, regarding the survival of OPSCC patients. First, we evaluated the prognostic impact of staging alone, and as expected, stage I and II patients had a longer survival than stage III and IV patients (*p* = 0.002, Figure 4A). As previously demonstrated in Table 4, the density of Langerhans cells is associated with the tumor stage of patients. Boxplots illustrated that CD1a+ cells tended to be recruited more in the ST of low-stage patients (*p* = 0.057, Figure 2B). Correlation with patient survival demonstrates that a high infiltrate of Langerhans cells in the ST is significantly associated with a better prognosis compared with a low density (*p* = 0.03, Figure 4C and Table 3).

As described in Table 6, a combination was established between these two clinical and immune variables according to their prognostic performance. When the eighth version of the TNM was created, high-stage p16+ patients were downgraded from one group to another due to their favorable prognosis. In the same manner, we observed that high-stage patients with initially poor survival had a significantly improved prognosis when they had a high density of Langerhans cells. Based on this observation, these patients were grouped with low-stage patients to create a prognostic score (low versus high). This score was tested in relationship to OS and showed a significantly better survival for patients with a low score compared with those with a high score (*p* < 0.0001, Figure 4D). This result highlights a higher significance than stage or CD1a density used separately.

### 3.4. Development of a Model Improving the Prediction of Overall Survival in OPSCC Patients

Finally, a Cox multivariate analysis was performed including p16 status, staging, CD1a ST and FoxP3 IT densities to assess their independent contributions to OS. This result shows that the four factors are dependent on each other (Table 7). Indeed, p16 status is included in TNMv8, and as we have shown, there are correlations between Treg and p16 as well as between Langerhans cells and stage.

However, the combination of clinical factors with immune cells, as described above, allows these two scores to be independent. Multivariate analysis revealed that the two combinations were significant prognostic factors for OPSCCs, with better separations and prognostic values than each variable separately (Table 8). Therefore, such a signature provides the best prognostic information for OPSCC patients.

## 4. Discussion

In recent years, it has been well accepted that the immune system plays a critical role in cancer development. Thus, many efforts have been intensified to identify new immune markers that could provide accurate predictive and prognostic information. In the TME, the active dialog between tumor and immune cells represents essential clinical information that should be integrated into the staging system of patients. Indeed, it has been recently demonstrated that the establishment of a new immune TNM based on TIL infiltration for low-stage tongue carcinoma patients provides additional prognostic information discriminating T1N0M0 from T2N0M0 patients and, therefore, improves their therapeutic management [32]. These combinations, commonly referred to as immunoscores, have been increasingly investigated for various cancers. The most accepted immune combination concerns colon cancers for which, in addition to the classical TNM, a clinical quantification of CD3+ and CD8+ cells is now routinely performed [28,33]. Furthermore, in breast cancers, an evaluation of TILs based on hematoxylin and eosin staining and morphology is highly recommended, as well as in non-small cell lung cancers where CD8+ and CD45RO+ lymphocyte markers appear to be promising candidates for improving TNM [5,34,35]. Indeed, it has been shown that their infiltration is a predictive factor of distant metastasis-free survival and OS [36]. This issue attracts increasing attention in head and neck cancers. Zhang et al., demonstrated that a scoring system evaluating the infiltration of CD3 and CD8 is of particular interest to ameliorate the TNM staging for HNSCCs [37]. Moreover, we recently proposed a new immunoscore combining CD68, CD8 and FoxP3 local distribution to identify patients with longer RFS and OS. Of note, this combination better discriminated HNSCC patients than the TNM classification [30].

In the current study, levels of CD8, FoxP3, CD68 and CD1a densities were assessed in both stromal and intratumoral compartments because it has been reported that their localization can result in different prognostic responses. As an example, Khoury et al., demonstrated that ST TILs can be distinguished from their IT counterparts regarding their biological behavior [27]. Hence, we determined immune cell density among a population of OPSCC patients, and based on the calculated cutoffs, we investigated their potential prognostic implications. Both IT CD68 and FoxP3 as well as ST CD1a correlated significantly with patient survival. Therefore, to identify differences in immune context between HPV-infected and -uninfected patients and between patients with low-stage and high-stage tumors, we compared the recruitment of immune cells between these four groups. Interestingly, we found that IT Treg infiltration was significantly different between p16+ and p16− patients, with greater infiltration observed in infected individuals. Similarly, stromal infiltrating Langerhans cells were also different between low- and high-stage patients, with a higher density of cells observed among low-stage tumors.

These four subgroups were specifically explored because they represent a challenge in terms of treatment. In fact, as the incidence of HPV+ OPSCC continues to rise, prognostic tools are sought to limit the deleterious side effects of surgery and radiotherapy techniques [38]. The aim of this principle of deintensification of treatments is to limit comorbidities and improve the quality of life and such approach is currently widely studied [39]. Although the eighth edition introduced a new classification for HPV+ tumors, we remain convinced that a method incorporating tumor biology, as represented by the immune system, would improve prognosis and patient management. Based on our promising results, we tested the prognostic performance of a score combining Treg density with tumor p16 status and demonstrated that this score provides a stronger discrimination than each parameter alone. Despite controversial findings in the literature, the prognostic role of Tregs remains frequently associated with improved survival in HNSCCs. Recently, a team made the same observations where FoxP3 was more highly expressed among HPV+ patients, and this high infiltration correlated with a better 5-year survival [40]. Moreover, we previously reported that FoxP3+ infiltration was associated with longer RFS and OS of patients suffering from HNSSC [15,16]. As we discussed previously, two populations of Tregs have been identified, one with immunosuppressive capabilities and the other with a proinflammatory role, both being associated with opposite prognoses [41]. Given the biological involvement of Tregs in HNSCCs and their positive prognostic impact, it seems relevant to consider this immune type in the risk stratification of such patients.

Quantification of the Langerhans cell number is not well documented for OPSCCs. They constitute a population of dendritic cells involved in antitumor immunity. Their main functions are to activate CD8+ lymphocytes, B lymphocytes and natural killer cells. These immature dendritic cells are characterized by the expression of the CD1a glycoprotein, the expression of which is reported to vary from one anatomical site to another. Indeed, we already reported that Langerhans cell infiltration was increased in HNSCCs compared with dysplastic lesions, whereas Gama-Cuellar et al., recently showed IT CD1a depletion in tonsillar carcinomas [24,42]. Moreover, other groups demonstrated in oral carcinomas that the decreased number of CD1a+ cells may be associated with cancer development [43,44]. Regarding their prognostic implications, the literature presents conflicting results. In the current study, we observed that high Langerhans cell density is synonymous with a better OS. This is in accordance with the findings of Karpathiou et al., who reported that high LC infiltration is associated with good prognostic values. Moreover, they demonstrated, like us, that a higher density of the dendritic cell marker S100 was associated with a lower T stage [45]. Additionally, a higher number of CD1a cells adjacent to the tumor improved the survival of tongue carcinoma patients [46]. In contrast, Minesaki et al., found that CD1a infiltration was an unfavorable prognostic factor for advanced laryngeal cancer [47].

Based on the reclassification performed on HPV+ OPSCC in 2018, we found with our score, combining staging with Langerhans cell density, that high-stage OPSCC highly infiltrated in ST had a better prognostic value than stage or LC infiltration alone. As proposed for p16 in the eighth edition of the AJCC, this group of high-stage patients could be downgraded when stromal-infiltrating CD1a+ cells are above the defined cutoff. Thus, we propose to clinically investigate their density in patients with T3 and T4 stages to better predict patient prognoses and to better guide the clinician in the treatment alternatives for these patients. However, a few limitations in our study need to be underlined, such as its retrospective design and the quantification of immune markers, which should be standardized based on digital scoring. Additionally, the number of cancer tissues that were used to examine immune cells should be increased. Nevertheless, even if they are low, the amount of immune information was strong enough to improve staging and p16 prognostic values, supporting the biological relevance of these combinations. Variables related to the TNM should also be collected in a comprehensive manner to accurately stratify low- and high-stage patients. Nevertheless, these encouraging results support the need for further studies focusing on FoxP3 and CD1a density and localization in OPSCC patients.

## 5. Conclusions

In conclusion, we have pointed out the importance of immune parameters as a part of the TNM staging system. The results reported in our work highlight the relevance of FoxP3 and CD1a densities in association with p16 positivity and TNM staging, respectively, to predict the prognosis of OPSCC patients with a greater accuracy. Indeed, these new combinations demonstrated a powerful prognostic value, outperforming p16 positivity and TNM systems alone. Ultimately, such new immune marker-containing scores should be implemented routinely to refine the prognostication and therapeutic management of these patients.

## Figures and Tables

**Figure 1 cancers-14-05560-f001:**
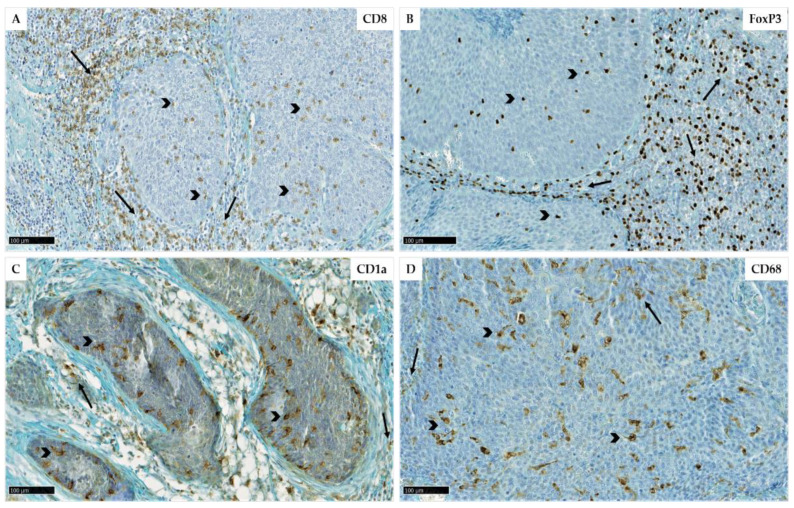
Immunohistochemical representation of CD8 (**A**), FoxP3 (**B**), CD1a (**C**) and CD68 (**D**) expression (scales = 100 µm) in stromal (arrows) and intratumoral (arrowheads) areas of oropharyngeal carcinomas. Bars = 100 µm.

**Figure 2 cancers-14-05560-f002:**
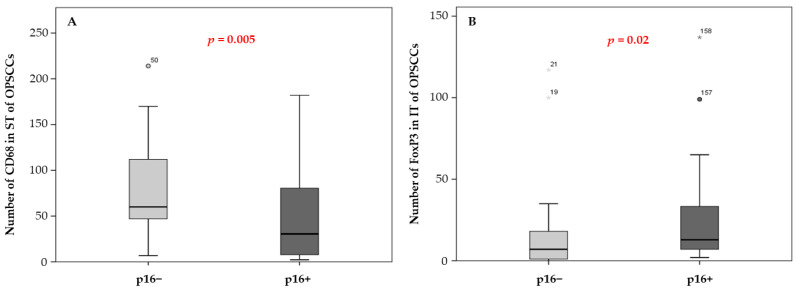
Evaluation of CD68 macrophage number in the ST compartment (**A**) and FoxP3 number in the IT area (**B**) of oropharyngeal tumors according to p16 status (Mann-–Whitney U test, *p* = 0.005 and *p* = 0.02, respectively) (*) asterisk symbols correspond to extreme atypical values.

**Figure 3 cancers-14-05560-f003:**
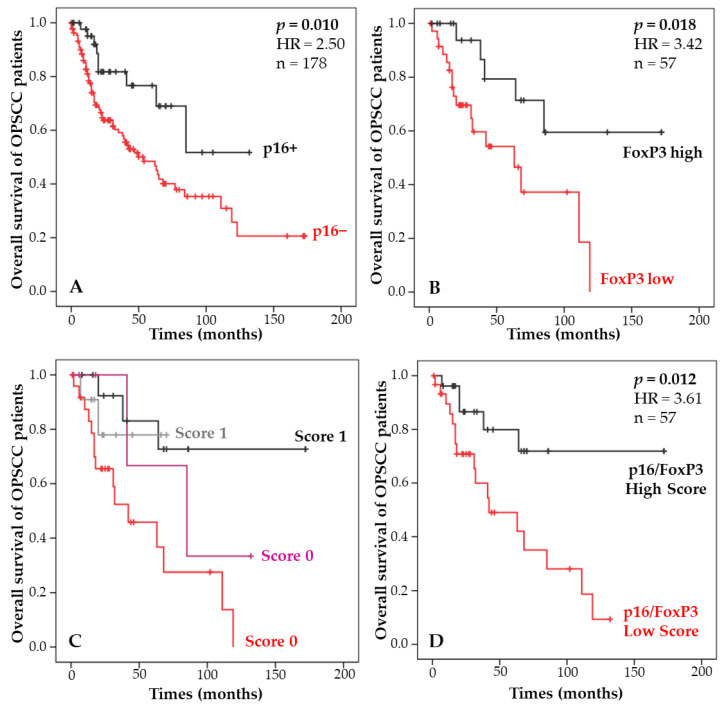
Kaplan-Meier curves of the OS of OPSCC patients according to p16 status (**A**), the number of FoxP3+ cells infiltrating the tumor (**B**), the combination of both parameters resulting in four groups (**C**), and the score combining p16 status and FoxP3 density (**D**).

**Figure 4 cancers-14-05560-f004:**
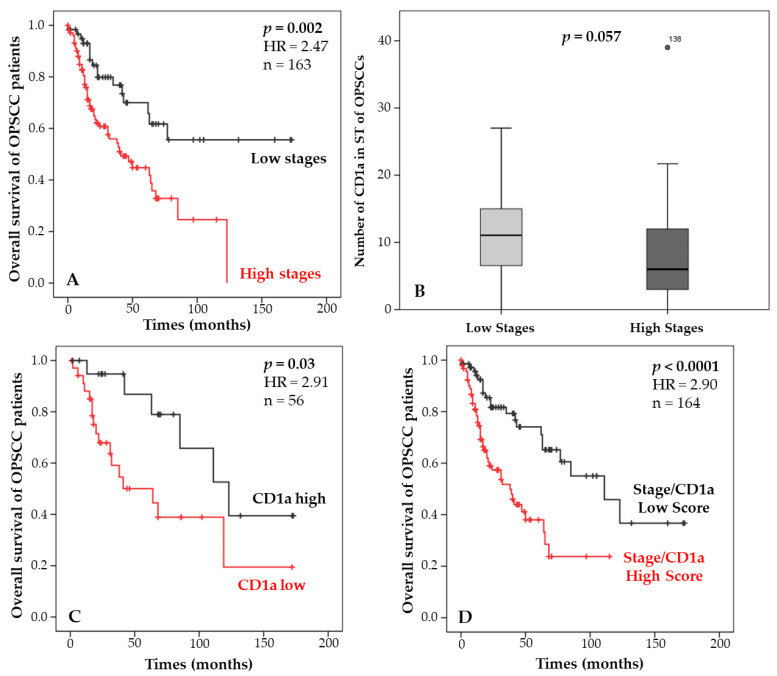
Kaplan–Meier curves of the OS of OPSCC patients according to staging (**A**), the number of CD1a+ cells infiltrating the ST (**C**), and the score combining the staging and CD1a density (**D**). Mann–Whitney test illustrating the number of CD1a cells in the ST of OPSCC patients according to the staging (low (I/II) versus high (III/IV) stages, as described in the TNM staging system 8) (**B**).

**Table 1 cancers-14-05560-t001:** Clinical patient characteristics.

Variables	Number of OPSCC Cases
	n = 194
**Age (years)**	
Median (range)	59 (24–89)
**Recurrence (RFS) (months)**	
Median (range)	17 (1–188)
Yes	75
No	106
Unknown	13
**Overall survival (OS) (months)**	
Median (range)	23 (1–173)
Alive	104
Dead	76
Unknown	14
**Gender**	
Male	130
Female	64
**Tumor stage 8th**	
I–II	67
III–IV	107
Unknown	20
**Histological grade**	
Undifferentiated	50
Poorly differentiated	53
Moderately differentiated	9
Well differentiated	56
Unknown	26
**Risk factors**	
**Tobacco**	
Smoker	147
Non-Smoker	35
Unknown	12
**Alcohol**	
Drinker	132
Non-Drinker	50
Unknown	12
**HPV detection**	
Positive	27
Negative	49
Unknown	118
**p16 staining**	
Positive	52
Negative	142
**p16 status**	
p16+	52
p16+/HPV−	0
p16−/HPV+	9
p16−/HPV−	133

**Table 2 cancers-14-05560-t002:** Descriptive table of immune cell density according to different subgroups including p16-negative, -positive, low-stage and high-stage patients.

Subgroups	Parameters	CD8 ST	CD8 IT	CD68 ST	CD68 IT	CD1a ST	CD1a IT	FoxP3 ST	FoxP3 IT
**All data**	n	44	44	77	77	66	66	69	69
	Median	90.6	14.4	55	16	8	55.5	92.5	12.6
	Min-Max	0–406.6	0–317.8	2.3–214	0–110	0–39	0–563	16–467	0–137
**p16-** **tumors**	n	24	24	49	49	52	52	45	45
	Median	90.6	13.35	60	14.1	8.35	53.5	88.2	7
	Min-Max	0–378.6	0–36.4	6.8–214	0–65	0–39	0–563	16–320	0–117
**p16+** **tumors**	n	20	20	28	28	14	14	24	24
	Median	94.35	19.85	30.5	18	6	66	93.35	12.9
	Min-Max	3.3–406.6	0.2–317.8	2.3–182	0–110	0–27	0.8–180	17–467	2–137
**Low stage patients**	n	21	21	35	35	24	24	29	29
	Median	90.2	17.7	47	16	11.05	29.5	105	11
	Min-Max	3.3–406.6	0.2–317.8	2.3–182	0–110	0–27	0.3–199	16–467	0–137
**High stage** **patients**	n	12	12	29	29	29	29	25	25
	Median	106.45	13.65	58	22	6	56	84.5	13
	Min-Max	7.2–196.9	0.2–57.6	6.5–214	0–68.6	0–39	0.2–563	17–362	0–117

**Table 3 cancers-14-05560-t003:** Univariate Cox regression analysis evaluating the influence of each immune cell type and their location (ST and IT) on OS. *p* values < 0.05 are highlighted in bold.

Univariate Analysis	Overall Survival	
	*p* Value	HR (95% CI)
CD8 ST	0.322	0.04 (0.0–25.7)
CD8 IT	0.173	0.35 (0.8–1.6)
CD68 ST	0.191	0.59 (0.3–1.3)
**CD68 IT**	**0.049**	**2.28 (1.0–5.2)**
FoxP3 ST	0.441	0.71 (0.30–1.68)
**FoxP3 IT**	**0.018**	**3.42 (0.10–0.81)**
**CD1a ST**	**0.029**	**2.91 (0.13–0.89)**
CD1a IT	0.183	0.54 (0.22–1.33)

**Table 4 cancers-14-05560-t004:** Mann–Whitney test between immune cell density in ST and IT compartments and p16 status or staging. *p* values < 0.05 are highlighted in bold.

Immune Cells	*p* Value versus p16	*p* Value versus Staging
CD8 ST	0.925	0.518
CD8 IT	0.071	0.868
CD68 ST	0.005	0.121
**CD68 IT**	**0.155**	0.761
FoxP3 ST	0.29	0.263
**FoxP3 IT**	**0.022**	0.627
**CD1a ST**	0.588	**0.057**
CD1a IT	0.451	0.335

**Table 5 cancers-14-05560-t005:** Description of the prognostic performance of a score combining p16 status and FoxP3 infiltration in OPSCC patients.

Curves	p16	FoxP3 IT	Survival	Score
Red	Negative	Low	Poor − −	Low
Black	Negative	High	Good ++	High
Gray	Positive	Low	Good +	High
Purple	Positive	High	Poor −	Low

**Table 6 cancers-14-05560-t006:** Description of the prognostic performance of a score combining the staging and CD1a infiltration in OPSCC patients.

Stage	Survival Stage	CD1a ST	Survival CD1a ST	Survival Using Combination	Score
Low	Good	Low	Poor	Good	Low
Low	Good	High	Good	Good	Low
High	Poor	Low	Poor	Poor	High
High	Poor	High	Good	Good	Low

**Table 7 cancers-14-05560-t007:** Multivariate analysis evaluating the correlation between p16 positivity, staging, stromal CD1a, intratumoral FoxP3 and patient survival. *p* values < 0.05 are highlighted in bold.

Multivariate Analysis	Overall Survival	
	*p* Value	HR (95% CI)
p16	0.974	1.03 (0.19–5.50)
Staging	**0.052**	3.20 (0.98–10.40)
CD1a ST	0.376	1.70 (0.52–5.53)
FoxP3 IT	0.057	3.15 (0.96–10.31)

**Table 8 cancers-14-05560-t008:** Multivariate analysis evaluating the correlation between the p16/FoxP3 score, stage/CD1a score and patient survival. *p* values < 0.05 are highlighted in bold.

Multivariate Analysis	Overall Survival	
	*p* Value	HR (95% CI)
p16/FoxP3 score	**0.038**	3.24 (1.06–9.85)
Stage/CD1a score	**0.032**	2.92 (1.09–7.77)

## Data Availability

Data is contained within the article or Appendix A.

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
