# Peer review of "Immune Cell Density Evaluation Improves the Prognostic Values of Staging and p16 in Oropharyngeal Cancer"

_cancers, 2022, doi:10.3390/cancers14225560_

Round 1
Reviewer 1 Report
The authors aimed to improve the classification system of OPSCC patients, regardless of HPV status, and to investigate the extent to which a combination of immune cells and clinical variables influence the prognosis.
The study covers some issues that have been overlooked in other similar topics. The structure of the manuscript appears adequate and well divided in the sections. Moreover, the study is easy to follow, but some issues should be improved. Some of the comments that would improve the overall quality of the study are:
a. Authors must pay attention to the technical terms acronyms they used in the text.
b. Conclusion Section: This paragraph required a general revision to eliminate redundant sentences and to add some "take-home message".
Author Response
The authors aimed to improve the classification system of OPSCC patients, regardless of HPV status, and to investigate the extent to which a combination of immune cells and clinical variables influence the prognosis.
The study covers some issues that have been overlooked in other similar topics. The structure of the manuscript appears adequate and well divided in the sections. Moreover, the study is easy to follow, but some issues should be improved. Some of the comments that would improve the overall quality of the study are:
Dear reviewer,
We would like to thank you for reviewing our paper and providing relevant comments that will help us to improve the quality of our publication. You will find below the detailing list of our corrections made in our article in response to your recommendations.
- Authors must pay attention to the technical terms acronyms they used in the text.
Response: Thank you for this comment. We have verified that each abbreviation was defined the first time it was cited. In addition, we have replaced some acronyms with the full name. Moreover, our manuscript was sent for a language editing by a US-trained expert.
- Conclusion Section: This paragraph required a general revision to eliminate redundant sentences and to add some "take-home message".
Response: We have reworked and simplified the conclusion (line 396) by mentioning only the major observations of our work.

Reviewer 2 Report
I read this article with great interest. The topic is of scientific soundness and the authors propose the basis of a new revision of TNM staging. The introduction is well written and essential, the methods are strong and correct and the results interesting as the discussion is. The figures and tables are clear and well descriptive. The article is suitable for publication
Author Response
I read this article with great interest. The topic is of scientific soundness and the authors propose the basis of a new revision of TNM staging. The introduction is well written and essential, the methods are strong and correct and the results interesting as the discussion is. The figures and tables are clear and well descriptive. The article is suitable for publication.
Dear reviewer,
We would like to thank you for taking the time to review our article and for highlighting its qualities.

Reviewer 3 Report
The manuscript examines expression of immunological markers CD8, FoxP3, CD1a, and CD68 in 194 human head and neck cancer cases. Intratumoral CD68+ macrophages, CD8+ T lymphocytes, and stromal CD1a+ Langerhans cells correlated with overall survival in univariate analyses. Stromal CD68 and intratumoral FoxP3 expression correlated with p16 status. High levels of intratumoral FoxP3+ regulatory T lymphocytes and stromal CD1a+ Langerhans cells correlated with overall survival and were observed in p16-positive and stage I-II patients. p16 and TNM staging were more efficient than clinical parameters alone to predict patient survival. The manuscript demonstrated that the combination of FoxP3 or CD1a with p16 status or staging were independent prognostic markers of outcome.
1. The p value of stromal CD1a expression and stage is over p<0.05; the significance of this result is uncertain.
2. Increased overall survival in stage I-II and p16+ head and neck cancer has been demonstrated by previous studies.
3. The manuscript requires grammatical, typographical, and style corrections.
Author Response
The manuscript examines expression of immunological markers CD8, FoxP3, CD1a, and CD68 in 194 human head and neck cancer cases. Intratumoral CD68+ macrophages, CD8+ T lymphocytes, and stromal CD1a+ Langerhans cells correlated with overall survival in univariate analyses. Stromal CD68 and intratumoral FoxP3 expression correlated with p16 status. High levels of intratumoral FoxP3+ regulatory T lymphocytes and stromal CD1a+ Langerhans cells correlated with overall survival and were observed in p16-positive and stage I-II patients. p16 and TNM staging were more efficient than clinical parameters alone to predict patient survival. The manuscript demonstrated that the combination of FoxP3 or CD1a with p16 status or staging were independent prognostic markers of outcome.
Dear reviewer,
We would like to thank you for reviewing our paper and providing relevant comments that will help us to improve the quality of our publication. You will find below the detailing list of our corrections made in our article (in Track Changes) in response to your recommendations.
- The p value of stromal CD1a expression and stage is over p<0.05; the significance of this result is uncertain.
Response: We agree with your comment. Therefore, we have corrected our statement in the results section, lines 227-228, by stating that the correlation was close to significance. Indeed, we showed a trend regarding a higher infiltration of Langerhans cells in low-stage patients, but the density of these cells alone correlated with the prognosis of OPSCC patients (p =0.03). It is for this reason that we made combinations between CD1a+ cell density and clinical stage, whose scores show a better significance (p <0.0001) for OS. However, we agree that the number of patients should be increased, and this was specified in the discussion section.
- Increased overall survival in stage I-II and p16+ head and neck cancer has been demonstrated by previous studies.
Response: Your comment is right, numerous studies reported the better prognosis of p16+ patients as well as low-stages patients with OPSCC. We have confirmed these better prognoses and have used this assumption to find new combinations that improve patient prognosis as well as classification.
- The manuscript requires grammatical, typographical, and style corrections.
Response: Our manuscript has been fully reviewed and sent to a platform for language editing by a US-trained expert.
